# Opportunistic Non-Governmental Organisation Delivery of a Virtual Stop Smoking Service in England during the COVID-19 Lockdown

**DOI:** 10.3390/ijerph19137722

**Published:** 2022-06-23

**Authors:** Nathan P. Davies, Matthew E. Callister, Harriet Copeland, Stuart Griffiths, Leah Holtam, Paul Lambert, Jacquelyn Mathur, Rebecca Thorley, Rachael L. Murray

**Affiliations:** 1School of Medicine, Nottingham City Hospital, University of Nottingham, Nottingham NG5 1PB, UK; nathan.davies@nottingham.ac.uk (N.P.D.); rebecca.thorley@nottingham.ac.uk (R.T.); 2Department of Respiratory Medicine, Leeds Teaching Hospitals, Leeds LS1 3EX, UK; matthew.callister@nhs.net (M.E.C.); harriet.copeland3@nhs.net (H.C.); 3Yorkshire Cancer Research, Jacob Smith House, 7 Grove Park Court, Harrogate HG1 4DP, UK; stuart.griffiths@ycr.org.uk (S.G.); leah.holtam@ycr.org.uk (L.H.); paul.lambert@ycr.org.uk (P.L.); jackie.mathur@ycr.org.uk (J.M.); 4SPECTRUM Consortium, Edinburgh EH8 9AG, UK

**Keywords:** tobacco, smoking, smoking cessation, COVID-19, service delivery, England

## Abstract

Smoking cessation services have rapidly transformed during the COVID-19 pandemic. Changes include pivoting from face-to-face to telephone and video call support, remote provision of stop smoking aids and more flexible appointments. This study reports an evaluation of a charity-led smoking cessation service rapidly conceived and launched in this context. The pilot service accepted self-referrals in Yorkshire, England from 20 May 2020 to 5 June 2020. A dedicated smoking cessation practitioner provided 12 weeks of weekly behavioural support over telephone or video call. NRT and/or medication and/or e-cigarettes were posted to the participant bi-weekly for up to 12 weeks. Written and telephone evaluation questionnaires were administered post-programme. Of 79 participants, 57 (72.2%) self-reported a 4-week quit and 51 (64.6%) self-reported a 12-week quit. Those concurrently using e-cigarettes and NRT had an 84.1% 12-week quit rate. The majority of participants chose to use e-cigarettes and NRT in combination (55.7%). 39 participants completed an evaluation form, with at least 90% recording they were “very satisfied” with each service component. 27 participants completed a telephone interview, reporting a relationship with practitioners, as well as convenience, and organisational reputation as service strengths. Virtual services can be set up quickly and effectively in response to demand. Quit rates were highest for those concurrently using e-cigarettes and NRT. Service users value flexibility and convenience of remote support and posting of quit aids.

## 1. Introduction

Smoking cessation services have faced both challenges and opportunities during the COVID-19 pandemic. Challenges have included the possible deterioration in smoking cessation programme outcomes [1], reduced opportunity for face-to-face contact and increased smoking amongst some groups of smokers [2,3,4]. Opportunities have included expansion of remote support and increased service flexibility [5,6] including mail delivery of medications [7] and increased motivation to quit in some populations [8].

### 1.1. Global Smoking Cessation Services in COVID-19 Pandemic

Expansion of remote services has been a key change for many services. A 2019 systematic review found moderate-certainty evidence that proactive telephone counselling helps smokers to quit smoking, but insufficient evidence to assess whether telephone counselling provided as an adjunct to other smoking cessation therapies has any additional effect [9]. There are promising signs that remote cessation services operated with some success in the pandemic; for example, studies required to change interventions from face-to-face behavioural support to remote behavioural support due to the pandemic found that quit rates [10] and clinic visits [11] for those in the intervention arm were similar for face-to-face and remote delivery. In Canada, it was found that the move to phone-based care for smoking cessation for cancer patients during the COVID-19 pandemic sometimes decreased counselling interruptions and improved follow-up rates [7].

### 1.2. English Smoking Cessation Services in COVID-19 Pandemic

In March 2020, the national training body in England recommended that all face-to-face smoking cessation provision be paused [12]. Action on Smoking and Health’s (ASH) annual survey of local smoking cessation services found that in August 2021, when national restrictions had been significantly relaxed, 17% of services still did not offer face-to-face support, whereas 98% of services offered telephone support and 60% were using real-time video support [5]. For context, no local services had been providing video support in August 2019 [13] and it is an intervention comparatively understudied [14]. Services also adapted their provision of nicotine replacement therapies (NRT), medication and e-cigarettes, by posting supplies, arranging home delivery and providing vouchers for purchase [5]. Local English services self-reported largely positive service user responses to these new ways of delivery [5,6], although managers and commissioners have been equivocal on the overall effect of the pandemic on service impact [5]. While there have been large cross-sectional surveys on the impact of COVID-19 on smoking cessation services, there are few in-depth service case studies.

### 1.3. Pilot Study

The Yorkshire Enhanced Stop Smoking Study (YESS) was set up alongside the Yorkshire Lung Screening Trial (YLST) to measure the effectiveness of a personalised smoking cessation service integrated with a lung screening programme.

Following the national lockdown introduced by the UK government in March 2020, both the YLST and YESS trials were paused for new recruitment. Rather than using the government furlough scheme for smoking cessation practitioners (SCP), in conversation with the funder of the trials, Yorkshire Cancer Research, the YESS trial team agreed to develop a short-term stop smoking pilot (“the service”) promoted by Yorkshire Cancer Research and delivered by the YESS study team. This single group retrospective evaluation sought to identify the viability of a rapidly implemented virtual stop smoking service. It reports results and lessons from the service, including systematic approaches to gathering quit data and service user feedback.

## 2. Materials & Methods

### 2.1. Participant Enrolment

The service was promoted by Yorkshire Cancer Research service primarily through organic posts and paid advertising on Facebook (Menlo Park, CA, USA) geo-targeted at those in Yorkshire, and those who had indicated an interest in smoking on Facebook, and secondarily through organisational communication channels.

The service was open for registration from 20 May 2020 to 5 June 2020 until service capacity was reached. Participants must have been over 18 years of age and resident in Yorkshire to be eligible to enrol in the service. No other inclusion criteria were applied.

### 2.2. Intervention

The YESS trial manager contacted all those who had registered their interest through the Yorkshire Cancer Research website, explained the service and confirmed they would like to make a supported quit attempt, and allocated willing participants to an SCP. SCPs then made contact within one working day. Participants could choose from a range of NRT/Champix and/or e-cigarettes to aid their quit attempt. An initial two-week supply of NRT/Champix and/or e-cigarettes were posted to the participant. If participants requested pharmacotherapies, a pre-screening questionnaire was completed by the SCP and sent to the GP for prescribing. The SCP and participant met over phone or video call one week after their initial meeting to initiate the quit process, or set a quit date. SCP weekly contact and NRT/e-cigarettes were then provided bi-weekly for a maximum of 12 weeks.

SCPs were trained to National Centre for Smoking Cessation and Training (NCSCT) standards and provided support in accordance with NHS, NCSCT and National Institute for Health and Care Excellence guidance standards and with agreement from Leeds Teaching Hospitals NHS Trust, the host organisation of the service.

### 2.3. Data Collection and Management

Participant data, including contact details, demographic information and service process and outcome data was secured on a password protected service database using data management methods outlined in the YESS study protocol [15]. Patients were contacted by the study manager who gave details of the service and use of personal data. The patients were then asked if they consented to be part of the study. Upon agreement they were referred to a stop smoking advisor who contacted them via the telephone to begin support.

### 2.4. Process Evaluation

All consenting participants who engaged with the service following a 4-week follow up were asked to complete an online survey on their experiences with the service.

Of those who consented to provide feedback, follow-up phone interviews were conducted with a purposive sample selected to be broadly representative of the participant population (Appendix A).

### 2.5. Cost

High level cost estimates (incorporating advertising/recruitment, postage, SCP salary, NRT and e-cigarette costs assuming a 12-week treatment period; but excluding estates and administrative costs) were calculated to provide an indicative cost of providing a similar service.

### 2.6. Analyses

Descriptive statistics have been used to characterise the sample. The primary outcome was self-reported cessation at 12 weeks (7 day point prevalent abstinence) and secondary outcomes include self-reported cessation at 4 weeks (7 day point prevalent abstinence), and quantitative and qualitative summaries of participant perceptions of the service.

## 3. Results

94 people registered interest in the service, of which 79 (84%) signed up to the service when contacted. Of this group, 58 (73.4%) participants were female and 21 were male, and 29 (36.7%) had a postcode in the 20% most deprived Lower Super Output Areas (LSOA) according to the 2019 Index of Multiple Deprivation (IMD). 42 (53.2%) found the service through Facebook. Of those who responded to the survey, the majority of participants reported smoking between 10 and 20 cigarettes per day (84.6%).

Of 79 participants, 57 (72.2%) achieved a 4-week quit, and 51 (64.4%) achieved a 12-week quit. 6 and 24 participants were lost to follow-up at 4 and 12 weeks, respectively, and were assumed to have not quit smoking. All participants received behavioural counselling, and 78 participants used one or more smoking aids. 44 (55.7%) concurrently used NRT and an e-cigarette, with those in this group having a 12-week quit rate of 84.1%. A minority of participants opted to use single smoking aids in addition to behavioural support, with 9 using only Champix (varenicline), 7 using only e-cigarettes and 13 using only NRT (Table 1).

39 participants completed the online service evaluation form, of which 30 (77%) were female, 16 were aged 50 to 59 (41%) and 24 (62%) were in employment.

Survey respondents were generally very positive about their service experience. For each service component, at least 90% of respondents rated themselves as “very satisfied” (Figure 1).

27 participants were selected for a telephone interview about their service experience. These participants were demographically representative of the overall service population; 70% were female and 63% were over 40. 93% of telephone interview participants had quit through the service.

Many respondents reported they had been thinking about stopping smoking for a while and the timeliness of the advertised call to action led them to register with the service. Some participants reported that the association with Yorkshire Cancer Research made the call to action stronger; this appeared to be related to the link with cancer and the professional reputation of the organisation.


*P26: ‘The reason why is that I knew that with it being Yorkshire Cancer Research—that you know what you’re doing in terms of helping people around cancer and avoiding cancer, and that because of that the support would be right and that I would get support, it would be professional help.’*
Female over 40, quit.

Respondents largely reported the service to be motivating and supportive, which was often driven by the strong personal bond with their SCP, the strength of which surprised some participants. The service was often compared favourably to smoking cessation services previously accessed. Again, the strength of relationship with the SCP was a core component of many comparisons, as was the accessibility and variety of stop smoking aids.


*P8: ‘…it’s good to have a stranger to help as they are really motivating when compared to my social circle. The SCP is non-judgemental and really encouraging...’*
Male under 40, did not quit.


*P10: ‘Through the GP I went on Champix—this service is much better. The GP seems to rush you through and get rid of you—whereas I’ve got the SCP’s phone number—I can call her if I need her. She rings me once a week and we have a lovely chat.’*
Male over 40, quit.


*P18: ‘This service is better than before. Used to go to an office in Rotherham. Better that I don’t have to go anywhere or get the bus, just do it on the phone.’*
Anonymous, quit.

A minority of participants made suggestions for service improvement, although there were no consistent themes. Evening SCP availability, group sessions, CO monitoring and additional electronic support were put forward as ideas for development.


*P11: ‘A bit more face to face would be better although I realise that during COVID this is not a possibility…’*
Female under 40, quit.

The estimated indicative cost of providing the service was calculated as £133.14 per 4-week quitter and £390.38 per 12-week quitter. Further details can be found in Appendix A.

## 4. Discussion

This study provides evidence that atypical providers (in this case a charity and research trial team) can set up virtual services quickly and relatively cheaply in response to external stimuli, with successes in recruitment and quit rate. The 4-week quit rate of 72.2% compares favourably to the England figure of 59% [16]; although the study population was entirely self-selecting, which may have supported higher quit rates, this is still higher than the England quit rate of 64% for the highest socio-economic group, those in managerial and professional occupations [16].

The telephone support was most used and well evaluated by participants. Very few participants took up the option of video calls; however, given that the service was offered in the earlier stages of the pandemic, it is possible that, as has been seen by local services [5], increasing confidence in video call technology would have seen greater uptake with time. Many participants identified building a relationship with a single allocated SCP was key to making a successful quit attempt, which may be more important than the choice of virtual platform [11].

Posting stop smoking aids directly to participants was viewed by both providers and participants as a convenient and effective way to ensure access, reflecting national findings [5,6]. The variety of stop smoking aids on offer was attractive to participants, with most selecting a combination of NRT and e-cigarettes. Participants identified an increasing variety of e-cigarette nicotine strength and flavour may support this offer. Lower uptake of single quit aids makes it difficult to compare quit rates across quit aids.

### Limitations

The service was run opportunistically with a small, regional, self-referred participant population recruited through non-targeted methods, which limits generalizability of findings. This group may have been more motivated than a typical smoking cessation service clientele. However, 36.7% of participants lived in postcodes found in the most deprived quintile in England, suggesting the service was accessible to those from lower socio-economic groups and had potential to promote health equity.

Quits were self-reported rather than carbon monoxide validated, which likely overestimates quit rates [17]. This was in line with COVID-19 recommendations from the national smoking cessation training body [12]. Given that this was a retrospective evaluation of a service, it has not been possible to formally compare our findings to other smoking cessation programmes, and future research should investigate the efficacy of such programmes. Finally, the calculated cost estimates are very broad and should be considered an indicative cost only.

Non-governmental organisations considering setting up smoking cessation services should be mindful of unintended impacts on local services. If “easier” quits from those with greater motivation and resources are diverted from local services, this can impact on performance indicators, in turn affecting funding and negatively impacting the ability to reach smokers with lesser opportunity to quit. Potential service providers should hold discussions with relevant stakeholders to ensure the new service contributes towards a decreasing overall population prevalence, not just supporting individual quits. These services also should ensure their data is included in regional and national data collection.

## 5. Conclusions

Virtual smoking cessation services can be set up quickly, effectively, and relatively cheaply by atypical providers in response to demand. Service users value the flexibility and convenience of remote support and more direct supply of quit aids. Atypical providers should consider how they integrate into the existing system of smoking cessation provisions in their area.

## Figures and Tables

**Figure 1 ijerph-19-07722-f001:**
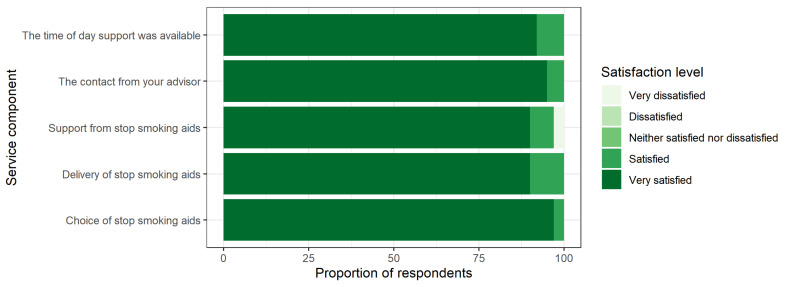
Service rating by category.

**Table 1 ijerph-19-07722-t001:** Summary of demographic information, entry pathway, reasons for accessing the service and sources of cessation support used by participants.

	Overall (*n* = 79)	4-Week Quitters (*n* = 57)	12-Week Quitters (*n* = 51)	4-Week Quitters (*n* = 57)	12-Week Quitters (*n* = 51)
		Within Category Review		Review by 4 and 12 Week Quits	
	*n* (%)	*n* (%)	*n* (%)	*n* (%)	*n* (%)
**All participants**	79 (100%)	57 (72.2%)	51 (64.6%)	57 (72.2%)	51 (64.6%)
**Age range**					
20–29	12 (15.2%)	10 (17.5%)	7 (13.7%)	10 (83.3%)	7 (58.3%)
30–39	17 (21.5%)	9 (15.8%)	6 (11.8%)	9 (52.9%)	6 (35.3%)
40–49	17 (21.5%)	12 (21.1%)	11 (21.6%)	12 (70.6%)	11 (64.7%)
50–59	22 (27.8%)	17 (29.8%)	16 (35.3%)	17 (77.3%)	16 (72.7%)
60–69	9 (11.4%)	7 (12.3%)	7 (13.7%)	7 (77.8%)	7 (77.8%)
70–79	2 (2.5%)	2 (3.5%)	2 (3.9%)	2 (100%)	2 (100%)
**Gender**					
Male	21 (26.6%)	15 (26.3%)	13 (25.5%)	15 (71.4%)	13 (61.9%)
Female	58 (73.4%)	42 (73.7%)	38 (74.5%)	42 (72.4%)	38 (65.5%)
**Support provided (in addition to behavioural counselling)**					
Champix	9 (11.4%)	8 (14.0%)	6 (11.8%)	8 (88.9%)	6 (66.7%)
E-cigarette	7 (8.9%)	5 (8.8%)	3 (5.9%)	5 (71.4%)	3 (42.9%)
NRT	13 (16.5%)	7 (12.3%)	4 (7.8%)	7 (53.8%)	4 (30.8%)
E-cigarette & NRT	44 (55.7%)	36 (63.2%)	37 (72.5%)	36 (81.8%)	37 (84.1%)
Champix & E-cigarette	1 (1.3%)	0 (0.0%)	0 (0.0%)	0 (0.0%)	0 (0.0%)
Behavioural support only	1 (1.3%)	1 (1.7%)	1 (2.0%)	1 (100%)	1 (100%)

## Data Availability

The data underlying this article cannot be shared publicly to protect the privacy of participants. The data will be shared on reasonable request to the corresponding author.

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
