# Peer review of "Opportunistic Non-Governmental Organisation Delivery of a Virtual Stop Smoking Service in England during the COVID-19 Lockdown"

_ijerph, 2022, doi:10.3390/ijerph19137722_

Round 1

Reviewer 1 Report

There are several studies that have been conducted during the pandemic in different countries, similar to this one. A more complete literature review should be attempted on tobacco use cessation and telemedicine. Multiple types of tobacco products should be considered. Recruitment appears to be one of convenience, localized, and the sample size is small, albeit this is a pilot study. The self-reported quit rate is extremely high. I don't think there was a way to establish the validity of self-reports of cessation nor any comparison group.   

Reviewer 2 Report

The manuscript submitted for publication by Davies et al., titled: "Opportunistic non-governmental organisation delivery of a virtual stop smoking service in England during COVID-19 lockdown" is a report on an interesting approach on a delivery of a virtual smoking service during COVID-19 related lockdown in Yorkshire England. 

The manuscript is well written and nicely organized, flows well and is easy to follow. 

The reviewer would like to offer the following points for consideration:

1. Consider presenting the research question and the principal hypothesis at the end of the introduction section.

2. The inclusion criteria appear fairly broad. In this sense how were the authors able to normalize/correct for confounding factors ? What were the exclusion criteria for study participation? 

3. Consider providing a discussion on where you think your results can be attributed to. for example it is mentioned in the narrative that the pilot achieved similar or better rates that those in England. Can the authors provide a potential explanation for that (theorize) and provide potentially relevant literature to support their thesis?

4. Consider providing a more thorough discussion on the interpretation of the obtained results and how those compare to similar studies/paradigms.

Reviewer 3 Report

General comment

The COVID pandemic has offered an opportunity to advance innovation in the delivery of health care interventions by using ICTs. Smoking cessation services are among those health interventions that may benefit more from this innovation. This study is therefore pertinent and relevant. My only concern is that e-cigarettes are neither medical devices, nor approved pharmacotherapy tested by pharmacokinetic and safety studies. The goal should be nicotine cessation since we are treating nicotine addiction.

Material & Methods

Authors should indicate the type of epidemiologic study that they carried out. Pilot is a study feature, not the study type. The study also explored the feasibility of SC virtual services, authors should also consider this.

Did authors applied for ethical approval for the study?  If not, explain why and describe briefly whether authors complied with health research ethical standards.

 Intervention

“SCP weekly contact and NRT/e-cig- 81 cigarettes were then provided bi-weekly for up to 12 weeks!

This means that the follow-up and behavior counselling lasted 12 weeks? Authors should indicate this in the manuscript

RESULTS

Instead of only reporting abstinence rates for EC+NRT, authors shoud briefly describe how many smokers using varenicle or NRT,-single or combined- achieved abstinence at 4 and 12 weeks. This should be written in the abstract also.  

Discussion

Authors should indicate the small and non-representative sample as a major limitation to generalization. Yes, it is an exploratory study but this should be highlighted.  Authors have been cautious not considering abstinence rates as a study main outcome . However, authors should highlight that smokers using varenicline or NRT achieved also high abstinence rates, especially those using varenicline: outcomes are similar to those of EC+NRT. Additionally, EC+NRT group does not achieve nicotine abstinence and will remain using e-cigarettes in the long term.  

Round 2

Reviewer 1 Report

The manuscript is a demonstration/shows promise of an online program. The sample is small and may be quite selective. The main difficulty is that there is no means of comparison to another program alternative, or even through use of a multiple baseline protocol. The authors at least need to state that in the Discussion Limitations.   

Author Response

Thank you for reading our revision and for your additional comments.

Comment: The main difficulty is that there is no means of comparison to another program alternative, or even through use of a multiple baseline protocol. The authors at least need to state that in the Discussion Limitations.   

Response: We have added to the discussion that explicitly recognises this limitation (lines 221-224)

Given that this was a retrospective evaluation of a service, it has not been possible to formally compare our findings to other smoking cessation programmes, and future research should investigate the efficacy of such programmes.

Reviewer 2 Report

The authors have made a reasonable effort in addressing reviewer’s comments.

Author Response

We are glad we addressed the comments. Thank you for suppporting the improvement of the paper.